# Is a History of Falling Related to Oral Function? A Cross-Sectional Survey of Elderly Subjects in Rural Japan

**DOI:** 10.3390/ijerph16203843

**Published:** 2019-10-11

**Authors:** Yoko Hasegawa, Nobuhide Horii, Ayumi Sakuramoto-Sadakane, Koutatsu Nagai, Takahiro Ono, Takashi Sawada, Ken Shinmura, Hiromitsu Kishimoto

**Affiliations:** 1Department of Dentistry and Oral Surgery, Hyogo College of Medicine, 1-1 Mukogawa-cho, Nishinomiya, Hyogo 663-8501, Japan; no-horii@hyo-med.ac.jp (N.H.); ayu.cherry.ayu@gmail.com (A.S.-S.);; 2Division of Comprehensive Prosthodontics, Graduate School of Medical and Dental Sciences, Niigata University, Niigata 951-8514, Japan; ono@dent.niigata-u.ac.jp; 3Department of Physical Therapy, School of Rehabilitation, Hyogo University of Health Sciences, 1-3-6 Minatojima, Chuo-ku, Kobe, Hyogo 650-8530, Japan; nagai-k@huhs.ac.jp; 4Hyogo Dental Association, 5-7-18 Yamamoto-dori, Chuo-ku, Kobe, Hyogo 650-0003, Japan; ahfb4801@bca.bai.ne.jp; 5Division of General Medicine, Department of Internal Medicine, Hyogo College of Medicine, 1-1 Mukogawa-cho, Nishinomiya, Hyogo 663-8501, Japan; ke-shimmura@hyo-med.ac.jp

**Keywords:** cross-sectional survey, elderly, falling, anxiety for falling, occlusal force, oral function, physical motor function

## Abstract

Background: Deteriorated physical function makes older adults prone to fall, and it is therefore known to prompt elders to require long-term care. In this regard, oral function can be related to the loss of motor function. This cross-sectional study assessed the oral factors that increase the risk of falling among older adults. Methods: We surveyed 672 self-reliant elderly individuals aged ≥65 years who were dwelling in a rural area. We assessed each subject’s risk of falling and any related anxiety. Oral-related conditions (number of teeth, occlusal support, masticatory performance, occlusal force, and tongue pressure) and physical motor functions (gait speed, knee extension force, and one-legged standing) were also assessed. Statistical analyses were performed using Mann-Whitney’s U-test, the *χ*^2^ test, and a logistic regression model. Results: In all subjects, 23% had a history of falling, while 40% had anxiety over falling. Both factors were significantly higher among female subjects, who also had slower gait speeds, and greater lateral differences in occlusion. The subjects with histories of falling were older, had impaired physical motor function, and exhibited a decrease in occlusal force and left/right occlusal imbalances. We recognized similar trends for anxiety about falling. Conclusions: These results revealed that the risk of falling might be lessened by maintaining healthy teeth occlusion and promoting healthy oral function.

## 1. Introduction

Falls present one of the most serious and costly problems associated with older adulthood. Such incidents also strongly impact both health and quality of life for elderly people. In fact, approximately one in three people aged 65 and above experience at least one fall annually, and 10–15% result in serious injury [1,2]. Accidents are the fifth leading cause of death among adults aged 65 years and older (after cardiovascular diseases, cancer, stroke, and respiratory causes) [3]. Falls account for two-thirds of all accidental deaths [4]. Decreased musculoskeletal strength in the limbs can make individuals more prone to falling. This is particularly known as a cause of falling among older adults who require caregiving. Falls are the main causes of accidental deaths among persons aged 65 years and older. Reports also indicate that older adults who fall once are at a higher risk of falling again. Fall prevention is thus important both socially and in terms of reducing medical costs.

Along with fall experience, excretory disorder, decreases in the frequency of going out, and depression, deteriorate oral function is now recognized as a risk related to caregiving needs among older adults [5]. The risk factors associated with falling include delayed reaction time, deteriorated skeletal muscle strength, decreased balance, and deteriorated gait function. This is evidence that declined physical function leads to an increase in fall frequency. Some reports suggest that skeletal muscle strength and oral function are related [6,7]. It is well known that when there is deterioration in physical equilibrium, this results in the decreased ability to perform activities of daily living (ADL) and the ability to stand without assistance (thereby increasing the risk of falling), and some reports also indicate that it is related to the loss of occlusal support [8,9], which is thought to be related to the occurrence of falls. Improvement in physical equilibrium has also been reported as a result of improved masticatory function [10]. Furthermore, occlusal force is known to strongly influence both masticatory performance and food intake [11], while tooth loss decreases occlusal force and thereby reduces masticatory performance [7,12]. However, few studies have quantitatively examined and detailed the characteristics of oral function (i.e., occlusal force and masticatory performance) among older adults with recent falling experiences. Furthermore, while deteriorated physical motor function is associated with falls, there is insufficient evidence for the relationship between physical motor function and oral function in older adults with histories of falling.

This study revealed the relationship between oral function and the risk of falling, focusing on remaining teeth occlusion. In addition, we tried to determine the intra-oral factors that increase the risk of falling among older adults living in the Tanba-Sasayama area.

## 2. Methods

This cross-sectional study was approved by the institutional review board of Hyogo College of Medicine (approval no. Rinhi 0342) and is part of the Frail Elderly in the Tamba Sasayama-Area (FESTA) study. Tamba-Sasayama City is situated in the mountainous region of Hyogo prefecture. Its main industry is farming and agriculture. The city is aging more than by the average in Japan, and it is an area that seems to have a large distribution of elderly people (31.4% were 65 or above as of September 2015). We recruited study participants by posting advertisements in local newspapers and by placing posters at the Hyogo Medical University Sasayama Medical Center.

### 2.1. Research Subjects

This study’s subjects were 672 adults aged 65 and older (i.e., 223 males and 449 females aged 72.8 ± 5.9, mean ± S.D.). These individuals participated in a joint medical and dental science study on independent older adults between June 2016 and December 2017. The participants were included based on the following criteria: participants were independent elderly individuals who required less than level 1 care based on the long-term care insurance system in Japan [13]. Exclusion criteria were participants suspected to have moderate to severe dementia (Mini-Mental State Examination score <20) [13], and those requiring level 2 or higher care based on the long-term care insurance system in Japan. If there was a suspicion of dementia in participants, their self-reported questionnaire was considered unreliable and they already often had problems with oral health [14]. Participants completed their own applications. Subjects were provided with full information about this study’s purpose and methods and were required to provide written consent before participating.

The data used in this study were anonymized. All data were masked for analysis. The authors did not have access to any participants’ personal information.

### 2.2. Evaluating the Risk of Falling

A self-administered questionnaire survey was conducted using the Kihon checklist, which consists of 25 questions designed to screen individuals requiring care prevention in Japan [15]. We deemed subjects who responded “Yes” to “Have you fallen during the past year?” (hereinafter, history of falls) as those with a risk of falling. We also investigated subjects who responded “Yes” to “Do you have a lot of anxiety about falling?” (hereinafter, fall anxiety).

### 2.3. Physical Component Assessments

We evaluated bodily composition by performing bio-electrical impedance analyses with Inbody770 (Inbody Japan, Tokyo, Japan). We assessed skeletal muscle mass in the limbs according to the Skeletal Muscle Mass Index (SMI) [16]. Each subject’s body mass index (BMI) was also calculated.

### 2.4. Motor Function Evaluations

We investigated the physical functions related to falling. Here, gait speed tests were performed by telling subjects to “walk at your normal speed”. This was done to analyze normal gait speed (m/sec) (hereinafter, gait speed). Acceleration and deceleration were noted at the start and end of this process, respectively. Participants walked a total of 10 m but were given 1 m both before and after this measured range to accelerate and decelerate [17].

Knee extension force measurements were conducted using the manual muscle strength meter (Mobie, Sakai Medical Co., Ltd., Tokyo, Japan). Subjects were instructed to sit squarely on a bed with their knee joints bent at 90-degree angles (the buttocks did not rise). Measurements were taken twice with the dominant leg to assess maximum torque. Here, we used the value obtained after dividing torque by bodyweight [18].

Subjects were instructed to stand on one leg with their eyes open (hereinafter, standing on one leg). We evaluated the time (in seconds) starting from the moment the subject placed their hands on their waist and raised their dominant leg off the floor until the hands came off the waist, the position of the foot changed, or part of the body other than the supporting leg touched the floor [19]. This measurement was performed twice for each subject, with the fastest times being used for analysis.

### 2.5. Oral Assessment

Subjects were seated on a reclining care chair to receive oral examinations under bright artificial lighting. We obtained data on the number of remaining teeth, denture usage condition, occlusal force, occlusal support, masticatory performance, and tongue pressure. The number of remaining teeth included tooth stumps and third molars.

Maximum occlusal force was measured at the left and right first molars using the Occlusal Force-Meter GM10 (Nagano Keiki, Tokyo, Japan) [7]. Total occlusal force was thus evaluated. If a subject was missing their first molar tooth, measurements were taken with the closest tooth so that both upper and lower teeth were active in the biting process. Subjects wearing dentures for daily life had measurements taken while wearing them. The left/right side balance of the occlusal force (hereinafter, occlusal balance) was determined using the following formula: Occlusal balance (%) = (Difference between left/right occlusal forces/Total occlusal forces on the left and right side) × 100

To assess masticatory performance, subjects were asked to chew gummy jelly 30 times before spitting it out. As such, masticatory performance was evaluated using a 10-point scale (i.e., 0–9 = min–max) [20]. Tongue pressure was measured using a balloon-probe type device (JMS tongue pressure meter, JMS, Hiroshima, Japan). Two maximum tongue pressure measurements were taken from each subject to determine mean values [21]. About the above two assessments, subjects wearing dentures for daily life had measurements taken while wearing them.

Occlusal support was assessed according to the Eichner classification [22] (Group A: Occlusal support in all four occlusal supporting zones; Group B: Occlusal support in one to three occlusal supporting zones; and Group C: No occlusal support).

### 2.6. Statistical Analysis

After testing for the normality and homogeneity of variance, all data were submitted to the non-parametric method. The relationship between history of falling and fall anxiety as well as that between motor function and oral function were evaluated using either Mann-Whitney’s U-test or the *χ*^2^ test. We then examined the relationship between physical function and oral function according to the Spearman correlation coefficient (i.e., the correlation between the statistically significant factors of history of falling and oral function). We performed logistic regression analyses (forced entry) using history of falling as the objective variable (Yes = 1, No = 0) to elucidate the factors related to falling. Here, measurement items were determined according to Mann-Whitney’s U-tests or *χ*^2^ tests (*p* < 0.1 were explanatory variables, while age and sex were control variables). Analyses were performed for all subjects and also according to sex (significance levels were set at 5%).

## 3. Results

Most participants were women (67%). Furthermore, 23% of all subjects had experienced a fall during the past year, while 40% had anxiety about falling (Table 1). Subjects with one of either history or anxiety about falling accounted for 39%, which indicated that falling was a familiar problem. Further, a history of falling and anxiety about falling were significantly more common among females.

Aside from knee extension force and occlusal force, there were no differences between men and women in terms of motor function and oral function, respectively. The number of remaining teeth were almost identical to those presented by an investigation conducted by the Ministry of Health, Labour and Welfare [23].

Although a statistically significant correlation was observed between motor function and oral function (excluding occlusal balance), there was no strong correlation (Table 2). Occlusal force/masticatory performance showed a greater correlation coefficient with knee extension force compared to other items.

### 3.1. Relationships between Falling Experience, Motor Function, and Oral Function

Participants with histories of falling were clearly older, had deteriorated motor function, deteriorated occlusal force, and uneven left/right occlusal balance compared to those with no such histories (Table 3).

Males showed no difference in motor function between those with a history of falling and those without. However, those with a history of falling often had significantly decreased measurements for items related to oral function. In other words, subjects with a history of falling had fewer teeth, decreased masticatory performance, and (although not significant) exhibited low tongue pressure and swallowing power. The results for women were different; those with a history of falling had decreased motor function and poorer occlusal balance with respect to oral function compared to those without a history of falling. Thus, subjects with a history of falling had significantly slower gait speeds, less endurance when standing on one leg, and poorer occlusal balance.

### 3.2. Relationships between Anxiety about Falling, Motor Function, and Oral Function

Subject motor function and oral function were related to anxiety about falling with nearly the same trend seen for a history of falling (Table 4). That is, subjects with anxiety about falling exhibited significantly diminished motor and oral functions. Most male subjects with anxieties about falling had significantly lower motor and oral functions across all evaluated items when compared to subjects with no anxieties about falling. Women did not exhibit any significant differences among the items related to motor or oral function regardless of falling anxiety.

### 3.3. Factors Affecting History of Falling

A logistic regression analysis revealed that sex, gait speed, and occlusal balance were significant explanatory variables affecting the history of falling among subjects (Table 5). Meanwhile the odds of occlusal balance were significant, but the confidence interval was 1.012 and the lower limit of the confidence interval is 1.003, suggesting that the impact of occlusal balance on falling was weak. Here, being a woman, having a slower gait speed, and/or experiencing uneven left/right occlusal balance may be associated with an increased risk of falling.

A logistic regression analysis performed on participant sex indicated that occlusal balance was a significant explanatory variable for men. This suggests that balanced left/right occlusal force reduces the risk of falling. We did not identify any statistically significant explanatory variables among women.

## 4. Discussion

Epidemiologic studies have reported on the risk factors related to falling. The global annual rate of falling among individuals aged 65 is between 28% and 35% [2], and elderly falls are a trigger for nursing care. The survey on falls for elderly people revealed the factors of falls, and a prompt response to preventable factors among these is important for maintaining the functioning of the elderly. It is also important for reducing social health care costs. The causes of falls are divided into internal and external (e.g., slight steps and slippery floor) factors. Internal factors are known to be due to age-related changes such as skeletal muscle strength, visual loss, loss of balance, and changes in the higher nervous system and sensory system. In the report of risk factors for falls, skeletal muscle weakness was the highest risk factor for fall risk [24]. Reduced skeletal muscle mass is known to be associated with decreased nutrition intake. In this cross-sectional survey, we inspected the hypothesis that decreased oral function is associated with decreased nutritional intake and decreased skeletal muscle mass and physical activity, resulting in an increased risk of falls.

This survey evaluated not only the incidence of falls but also anxiety about falls, because the reports on anxiety about falling have also indicated that it is a risk factor for falling [25].

Our findings showed that a decrease in gait speed increases the risk of falling. Decreased gait speed has often been considered the strongest risk factor for adverse events like falling among older adults; it is also one of the most useful indicators for determining physical frailty [26,27]. It is also true that decreased lower-limb muscle strength and less endurance when standing on one leg are risk factors for falling [28,29].

This study’s results suggest a correlation between oral function and history of falling. In addition, older adults with poor occlusal balance are at high risk of falling, while elderly men with fewer teeth and lower masticatory performance tend to have more frequent histories of falling and greater associated fear. Furthermore, while there was no clear correlation between history/fear of falling and oral function (other than occlusal balance) among female subjects, they were generally at a greater risk of falling than male subjects. Women report more falls and experience more fall-related injuries than men [25,30]. While women tend to have more gait variability (variation in steps) and are therefore more likely to fall than men [31], elderly women have the additional risk of advanced post-menopausal osteoporosis and fractures, suffering during falls, and decreased ADL and quality of life. Particular efforts are thus needed to prevent falls among these individuals.

This study’s authors previously reported a correlation between occlusal balance and standing on one leg [32,33]. Kimura et al. further reported that time standing on one leg can be prolonged through dental treatment [34], while Ringhof et al. showed that occlusal control can stabilize normal posture and balance when standing on one leg [35]. Kimura et al. also reported that occlusal support while wearing dentures was related to Timed Up and Go and standing on one leg balance test results [34]. Thus, occlusal balance affects the balance of the body trunk in addition to lower-limb muscle strength and balance. Improvements to these areas therefore reduce the risk of falling. For this reason, it is necessary to restore oral function among older adults with significant left/right occlusal force imbalances.

This study was a cross-sectional survey and did not clarify a causal relationship with falling. The survey population mainly consisted of women. In addition, while occlusal force was assessed on the left and right first molars or neighboring teeth of each participant, occlusal force balance was not evaluated in a strict sense. Fall history in this study was confirmed through self-reports during the past year, so that self-reports have limited accuracy or reliability.

## 5. Conclusions

This study’s results suggest that it is important to prevent falling accidents not only by slowing the deterioration of motor function, but also by promoting and maintaining healthy masticatory performance and occlusal force. Dental treatment may also be beneficial for at-risk individuals with poor left/right occlusal balance. Furthermore, while anxiety about falling was associated with decreased exercise, it was also significantly associated with deteriorated oral function (e.g., occlusal force, tongue pressure, and masticatory performance). Thus, efforts to improve oral function among older adults may also prevent falling and remove falling-related anxieties.

## Figures and Tables

**Table 1 ijerph-16-03843-t001:** Subject characteristics.

Variables	Overall (672 Subjects)	Male (223 Subjects)	Female (449 Subjects)	*p*-Value
Age (years)	72.8 ± 5.9	73.2 ± 6.1	72.6 ± 5.8	0.21
BMI	22.5 ± 2.9	23.1 ± 2.7	22.2 ± 2.9	0.003
SMI	6.4 ± 0.9	7.4 ± 0.7	6 ± 0.6	<0.001
**Questionnaire for falling**				
Has a history of falling	151 (22.5)	38 (17)	113 (25.2)	0.02
Has anxiety about falling	271 (40.3)	62 (27.8)	209 (46.5)	<0.001
**History of and anxiety about falling**				
Has both history and anxiety	79 (11.8)	12 (5.4)	67 (14.9)	<0.001
Has one of either history or anxiety	264 (39.3)	76 (34.1)	188 (41.9)	
**Motor function**				
Gait speed (m/s)	1.5 ± 0.2	1.5 ± 0.2	1.5 ± 0.2	0.84
Knee extensor force (N·m/kg)	356.7 ± 120.5	459.5 ± 118.5	306 ± 83.7	*p* < 0.001
Standing on one leg (s)	27.6 ± 24.8	29.7 ± 27.1	26.5 ± 23.5	0.18
**Oral assesment**				
Number of teeth	20 ± 8.9	19.6 ± 9.4	20.3 ± 8.6	0.83
Wearing denture	318 (47.3)	111 (49.8)	207 (46.1)	0.21
Occlusal force (kgf)	59 ± 37	66 ± 42.6	55.6 ± 33.4	0.01
Occlusal balance (%)	23.8 ± 21.2	26 ± 22.5	22.6 ± 20.5	0.06
Mastication performance (score)	3.9 ± 2.3	4 ± 2.4	3.8 ± 2.2	0.32
Tongue pressure (kPa)	31.1 ± 8.7	31.7 ± 9	30.8 ± 8.6	0.16
Occlusal support				
Group A	292 (43.5)	93 (41.7)	199 (44.3)	0.23
Group B	268 (39.9)	85 (38.1)	183 (40.8)	
Group C	112 (16.7)	45 (20.2)	67 (14.9)	

*p*-Value: Comparison between male and female. Mann-Whitney’s U-test or *χ*^2^ test. SMI: Skeletal Muscle Mass Index Fall history: Subjects who responded “Yes” to “Have you fallen during the past year?”. Fall anxiety: Subjects who responded “Yes” to “Do you have a lot of anxiety about falling?”. Occlusal support: Eichner classification.

**Table 2 ijerph-16-03843-t002:** Relationship between oral assessments and motor functions.

Variables	Gait Speed		Knee Extension Force		Standing on One Leg	
Overall	*r*	*p*-Value	*r*	*p*-Value	*r*	*p*-Value
Number of teeth	0.161 **	<0.001	0.189 **	<0.001	0.189 **	<0.001
Occlusal force	0.201 **	<0.001	0.322 **	<0.001	0.198 **	<0.001
Occlusal balance	−0.002	0.968	0.018	0.646	−0.061	0.117
Masticatory performance	0.193 **	<0.001	0.235 **	<0.001	0.182 **	<0.001
Tongue pressure	0.07	0.07	0.216 **	<0.001	0.149 **	<0.001
Occlusal support	−0.144 **	<0.001	−0.172 **	<0.001	−0.192 **	<0.001
**Male**						
Number of teeth	0.162 *	0.016	0.242 **	<0.001	0.134 *	0.047
Occlusal force	0.237 **	<0.001	0.284 **	<0.001	0.188 **	0.005
Occlusal balance	−0.035	0.61	0.042	0.533	0.056	0.409
Masticatory performance	0.248 **	<0.001	0.272 **	<0.001	0.187 **	0.006
Tongue pressure	0.12	0.074	0.343 **	<0.001	0.188 **	0.005
Occlusal support	−0.113	0.095	−0.252 **	<0.001	−0.119	0.08
**Female**						
Number of teeth	0.160 **	0.001	0.224 **	<0.001	0.223 **	<0.001
Occlusal force	0.191 **	<0.001	0.335 **	<0.001	0.196 **	<0.001
Occlusal balance	0.01	0.828	−0.048	0.313	−0.130 **	0.006
Masticatory performance	0.167 **	<0.001	0.259 **	<0.001	0.179 **	<0.001
Tongue pressure	0.045	0.337	0.173 **	<0.001	0.126 **	0.008
Occlusal support	−0.162 **	0.001	−0.235 **	<0.001	−0.237 **	<0.001

*r*: Spearman correlation coefficient, **: *p* < 0.001, *: *p* < 0.05.

**Table 3 ijerph-16-03843-t003:** The relationships between falling experience, motor function, and oral function.

Overall (672 Subjects)	Yes (151 Subjects)	No (521 Subjects)	*p*-Value
Age (years) *	73.6 ± 6	72.5 ± 5.9	0.04
BMI	22.7 ± 2.8	22.5 ± 2.9	0.16
SMI	6.4 ± 0.9	6.4 ± 1	0.54
Motor function			
Gait speed (m/s) *	1.4 ± 0.2	1.5 ± 0.2	0.009
Knee extension force (N·m/kg) *	334.5 ± 113.4	363.1 ± 121.8	0.004
Standing on one leg (s) *	23.2 ± 21.6	28.8 ± 25.5	0.006
Oral function test			
Number of teeth	18.9 ± 9.3	20.4 ± 8.8	0.06
Wearing denture	79 (52.3)	239 (45.9)	0.096
Occlusal force (kgf) *	53.1 ± 34.9	60.8 ± 37.5	0.03
Occlusal balance (%) *	29 ± 25.6	22.2 ± 19.5	0.007
Masticatory performance (score)	3.5 ± 2.4	4 ± 2.2	0.056
Tongue pressure (kPa)	30 ± 8.9	31.4 ± 8.6	0.08
Occlusal support			
Group A	55 (36.4)	237 (45.5)	0.14
Group B	68 (45)	200 (38.4)	
Group C	28 (18.5)	84 (16.1)	
Male (223 subjects)	Yes (38 subjects)	No (185 subjects)	*p*-Value
Age (years)	74.1 ± 6.5	73 ± 6	0.35
BMI	22.8 ± 2.9	23.2 ± 2.7	0.44
SMI	7.3 ± 0.8	7.4 ± 0.7	0.53
Motor function			
Gait speed (m/s)	1.4 ± 0.2	1.5 ± 0.2	0.07
Knee extension force (N·m/kg)	440.1 ± 129.7	463.4 ± 116.1	0.11
Standing on one leg (s)	27.8 ± 23.5	30.1 ± 27.8	0.63
Oral function test			
Number of teeth *	17.1 ± 9.4	20.1 ± 9.3	0.03
Wearing denture	24 (63.2)	87 (47.0)	0.051
Occlusal force (kgf)	57 ± 37.8	67.9 ± 43.4	0.16
Occlusal balance (%) *	37.3 ± 30.4	23.7 ± 19.8	0.017
Masticatory performance (score) *	3.3 ± 2.3	4.1 ± 2.4	0.04
Tongue pressure (kPa)	29.6 ± 7.4	32.1 ± 9.2	0.07
Occlusal support			
Group A	11 (28.9)	82 (44.3)	0.21
Group B	18 (47.4)	67 (36.2)	
Group C	9 (23.7)	36 (19.5)	
Female (449 subjects)	Yes (113 subjects)	No (336 subjects)	*p*-Value
Age (years) *	73.4 ± 5.8	72.3 ± 5.8	0.045
BMI *	22.7 ± 2.7	22.1 ± 3	0.013
SMI	6 ± 0.6	5.9 ± 0.7	0.18
Motor function			
Gait speed (m/s) *	1.4 ± 0.2	1.5 ± 0.2	0.047
Knee extension force (N·m/kg)	300 ± 82.7	308.1 ± 84	0.21
Standing on one leg (s) *	21.7 ± 20.8	28.2 ± 24.1	0.005
Oral function test			
Number of teeth	19.5 ± 9.2	20.5 ± 8.4	0.43
Wearing denture	55 (48.7)	1527 (45.20)	0.30
Occlusal force (kgf)	51.8 ± 33.9	56.9 ± 33.2	0.13
Occlusal balance (%) *	26.3 ± 23.3	21.4 ± 19.3	0.04
Masticatory performance (score)	3.6 ± 2.4	3.9 ± 2.1	0.39
Tongue pressure (kPa)	30.2 ± 9.3	31 ± 8.3	0.41
Occlusal support			
Group A	44 (38.9)	155 (46.1)	0.41
Group B	50 (44.2)	133 (39.6)	
Group C	19 (16.8)	48 (14.3)	

Variables are the same as in Table 1. Results are shown for all subjects. *p*-Value: Comparison between with/without fall history. Mann-Whitney’s U-Test or *χ*^2^. *: There was a significant difference between those with and without fall history.

**Table 4 ijerph-16-03843-t004:** The relationship between anxiety over falling, motor function, and oral function.

Overall (672 Subjects)	Yes (271 Subjects)	No (401 Subjects)	*p*-Value
Age (years) *	73 ± 6	70 ± 5.7	<0.001
BMI	25.7 ± 2.7	21.6 ± 3	0.86
SMI *	5.5 ± 0.9	5.9 ± 1	<0.001
Motor function			
Gait speed (m/s) *	1.4 ± 0.3	1.5 ± 0.2	0.029
Knee extension force (N·m/kg) *	329.2 ± 104	375.3 ± 127.3	<0.001
Standing on one leg (s) *	25.4 ± 24.5	29 ± 24.8	0.017
Oral function test			
Number of teeth *	19.1 ± 9.2	20.7 ± 8.6	0.03
Wearing denture *	176 (52.2)	141 (43.9)	0.02
Occlusal force (kgf) *	52.1 ± 34	63.7 ± 38.2	<0.001
Occlusal balance (%)	24 ± 23	23.6 ± 19.9	0.48
Masticatory performance (score) *	3.7 ± 2.2	4 ± 2.3	0.044
Tongue pressure (kPa) *	29.8 ± 8.6	32 ± 8.7	0.003
Occlusal support			
Group A	109 (40.2)	183 (45.6)	0.096
Group B	107 (39.5)	161 (40.1)	
Group C	55 (20.3)	57 (14.2)	
Male (223 subjects)	Yes (62 subjects)	No (161 subjects)	*p*-Value
Age (years) *	75.2 ± 6.4	72.4 ± 5.8	0.002
BMI	23 ± 2.4	23.1 ± 2.8	0.94
SMI	7.4 ± 0.6	7.4 ± 0.7	0.89
Motor function			
Gait speed (m/s) *	1.4 ± 0.3	1.5 ± 0.2	0.04
Knee extension force (N·m/kg) *	431.1 ± 103.1	470.3 ± 122.4	0.03
Standing on one leg (s) *	23.6 ± 22.2	32.1 ± 28.5	0.02
Oral function test			
Number of teeth	17.4 ± 10.2	20.5 ± 9	0.054
Wearing denture *	37 (59.7)	74 (46.0)	0.046
Occlusal force (kgf) *	50.9 ± 36.7	71.9 ± 43.4	0.001
Occlusal balance (%)	30.6 ± 27.3	24.3 ± 20.2	0.31
Masticatory performance (score) *	3.5 ± 2.4	4.2 ± 2.4	0.047
Tongue pressure (kPa) *	29.1 ± 9.1	32.7 ± 8.7	0.009
Occlusal support *			
Group A	22 (35.5)	71 (44.1)	0.02
Group B	20 (32.3)	65 (40.4)	
Group C	20 (32.3)	25 (15.5)	
Female (449 subjects)	Yes (209 subjects)	No (240 subjects)	*p*-Value
Age (years) *	73.6 ± 5.9	71.7 ± 5.6	<0.001
BMI	22.4 ± 2.8	22.1 ± 3	0.32
SMI	5.9 ± 0.6	6 ± 0.7	0.21
Motor function			
Gait speed (m/s)	1.5 ± 0.3	1.5 ± 0.2	0.18
Knee extension force (*N*ηm/kg)	299.3 ± 83.4	311.9 ± 83.7	0.13
Standing on one leg (s)	26 ± 25.2	27 ± 21.9	0.24
Oral function test			
Number of teeth	19.7 ± 8.8	20.8 ± 8.4	0.15
Wearing denture	104 (50.0)	102 (42.5)	0.07
Occlusal force (kgf)	52.5 ± 33.3	58.3 ± 33.3	0.055
Occlusal balance (%)	22 ± 21.3	23.2 ± 19.8	0.3
Masticatory performance (score)	3.7 ± 2.1	3.9 ± 2.2	0.36
Tongue pressure (kPa)	30 ± 8.4	31.4 ± 8.7	0.1
Occlusal support			
Group A	87 (41.6)	112 (46.7)	0.45
Group B	87 (41.6)	96 (40)	
Group C	35 (16.7)	32 (13.3)	

Variables are the same as in Table 1. Results are shown for all subjects. *p*-Value: Comparison between with/without fall anxiety. Mann-Whitney’s U-test or *χ*^2^. *: There was a significant difference between those with and without fall history.

**Table 5 ijerph-16-03843-t005:** Factors affecting fall history.

History of Falls		*B*	Wald	*p*-Value	Odds Ratio	Confidence Interval	
						Lower	Upper
Overall	Age	0.005	0.058	0.810	1.005	0.967	1.043
	Sex *	0.568	4.176	0.041	1.764	1.024	3.040
	Gait speed *	−0.910	4.581	0.032	0.403	0.175	0.926
	High knee extension	0.000	0.021	0.885	1.000	0.998	1.002
	Standing on one leg	−0.007	2.037	0.153	0.993	0.984	1.002
	Number of teeth	0.002	0.007	0.933	1.002	0.967	1.037
	Wearing denture	0.063	0.054	0.816	1.065	0.629	1.804
	Occlusal force	0.001	0.091	0.763	1.001	0.993	1.009
	Occlusal balance *	0.012	7.275	0.007	1.012	1.003	1.022
	Masticatory performance	−0.026	0.165	0.685	0.975	0.862	1.103
	Tongue pressure	−0.009	0.549	0.459	0.991	0.969	1.014
	Constant	−0.627	0.105	0.746	0.534		
Male	Age	−0.003	0.010	0.920	0.997	0.933	1.064
	Gait speed	−1.066	1.646	0.200	0.344	0.068	1.755
	Number of teeth	0.008	0.057	0.812	1.008	0.946	1.074
	Wearing denture	0.316	0.374	0.541	1.372	0.498	3.784
	Occlusal balance *	0.019	5.639	0.018	1.019	1.003	1.035
	Masticatory performance	−0.037	0.109	0.742	0.964	0.773	1.202
	Tongue pressure	−0.015	0.437	0.509	0.985	0.943	1.029
	Constant	−0.090	0.001	0.978	0.914		
Female	Age	0.02	1.56	0.21	1.02	0.99	1.07
	BMI	0.08	3.67	0.06	1.08	1.00	1.16
	Occlusal balance	0.009	3.13	0.08	1.01	1.00	1.02
	Constant	−4.86	8.32	0.004	0.01		

History of fall was the objective variable (Yes = 1, No = 0), variables in Table 3 which the result of Mann-Whitney’s U-test or *χ*^2^ test were *p* < 0.1 were explanatory variables (For occlusal support, Group A was a reference variables), and sex and age were control variables. *: Statistically significant explanatory variables.

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
