# Peer review of "Is a History of Falling Related to Oral Function? A Cross-Sectional Survey of Elderly Subjects in Rural Japan"

_ijerph, 2019, doi:10.3390/ijerph16203843_

Round 1

Reviewer 1 Report

Consider clarifying inclusion & exclusion criteria;

Discuss limitations of self-reporting, especially in older patients with un/diagnosed dementia;

Consider further discussing link between mental & oral health -example: Delwel, S., Binnekade, T. T., Perez, R. S., Hertogh, C. M., Scherder, E. J., & Lobbezoo, F. (2018). Oral hygiene and oral health in older people with dementia: a comprehensive review with focus on oral soft tissues. Clinical oral investigations, 22(1), 93-108.

Was oral hygiene/ plaque-bleeding scores assessed?

Was salivary function (flow/buffering capacity) investigated?

Further clarify the link between occlusal balance and falling, especially with older patients possibly with a shortened dental arch/ multiple missing teeth - invariably having an impact on their occlusal scheme.

Author Response

Consider clarifying inclusion & exclusion criteria;

-       Thank you for your review. Based on your comments, we have clarified the inclusion and exclusion criteria of the method to make it easier to understand (Page 1, Lines 78 to Page 2, Lines 1).

Discuss limitations of self-reporting, especially in older patients with un/diagnosed dementia;

-       Based on your comments, we have added a limitation about self-reporting (Page 10, Line 255-256). Regarding this survey, the history of falls was judged using self-reports; we did not verify the validity or reproducibility of the judgement method as it was based on self-reporting. In a survey with a large sample size, it is practically impossible for an expert to make a detailed assessment of each participant. On the other hand, the participants of this study were independent older adults, and those with cognitive decline were excluded because the exclusion criteria to participate included a Mini-Mental State Examination score <20. Therefore, we suspect that the self-reporting in this survey was highly reliable.

Consider further discussing link between mental & oral health -example: Delwel, S., Binnekade, T. T., Perez, R. S., Hertogh, C. M., Scherder, E. J., & Lobbezoo, F. (2018). Oral hygiene and oral health in older people with dementia: a comprehensive review with focus on oral soft tissues. Clinical oral investigations, 22(1), 93-108.

-   Thank you for pointing this out. The participants of this study are only those whose Mini-Mental State Examination score determined that there was no problem with cognitive function. With reference to the information from the presented literature, we have added a description showing that there is a relationship between oral health and cognitive function and have revised the Material and Method section (Page 3, Lines 84).

Was oral hygiene/ plaque-bleeding scores assessed?

-    This study assessed the relationship between physical motor function and oral function, so oral hygiene/plaque-bleeding was not discussed. Because our survey is ongoing and we have collected data about oral hygiene status, we will be able to discuss this topic in a future study.

Was salivary function (flow/buffering capacity) investigated?

-        This study assessed the relationship between physical motor function and oral function, so salivary function (flow/buffering capacity) was not discussed. Because our survey is ongoing and we have collected data about dry-mouth status, we can consider this topic in a future study.

Further clarify the link between occlusal balance and falling, especially with older patients possibly with a shortened dental arch/ multiple missing teeth - invariably having an impact on their occlusal scheme.

-        In this study, participants were not classified based on shortened dentition, but were classified by occlusal support using Eichner classification [21].
-        Based on your suggestion, we conducted additional statistical analysis on the following items: Examination 1.The relationship between the number of remaining teeth and occlusal balance; and Examination 2. The relationship between the number of occlusal support areas of molars and occlusal balance. The participants of Eichner B can be divided into four subgroups according to the number of occlusal support areas in their molars (three, two, one, and zero location).
-        The results of Ex. 1, regarding the number of remaining teeth and occlusal balance, showed a weak negative correlation between the number of remaining teeth and occlusal balance (R=-0.23, P<0.001; Spearman’s rank correlation coefficient). The results of the examination of each of the three groups with occlusal support showed a significant weak negative correlation in the Eichner A group (R = -0.21, P <0.001; Spearman’s rank correlation coefficient). On the other hand, in the Eichner B and C groups, there was no significant correlation between the number of remaining teeth and occlusal balance.
-        The results of Ex. 2, showing the relationship between the number of occlusal support areas of molars and occlusal balance, indicated that there was no significant difference between the four subgroups of Eichner B (Kruskal Wallis test, P=0.558, the lower figure).

-        One reason for these results may be that the participants who wear dentures regularly were evaluated while wearing dentures.

-        We added the relationship between falling experience/anxiety over falling and denture wearing in a univariate analysis (in Tables 3 and 4) and a multivariate analysis (in Table 5). There was no significant association between denture wearing and falling experience/anxiety over falling.
-        The main topic of this study was to clarify the oral function factors related to falls in daily life, so the occlusal balance was evaluated while dentures were being worn, and we clarified the effect of falling on denture wearing in the results (Tables 3-5) .

Reviewer 2 Report

This article reported the association of the risk of falling and any related anxiety with oral-related conditions (number of teeth, occlusal support, masticatory performance, occlusal force, and tongue pressure) and physical motor functions (gait speed, knee extension force, and one-legged standing) in the elderly individuals. By a logistic regression model, some oral related conditions such as a decrease in occlusal force and left/right occlusal imbalances were related to physical motor function leading to an increase of the falling risk and anxiety of falling. In addition, the author analysed the association by sex.

The thesis of study was evaluated to be timely at the increase of the elderly population and health promotion for the elderly. The study employed the proper statistical tests. The conclusions of the study were derived from the study results.

However, some minor corrections are required. Bold expressions of table 3, table 4 and table 5 are not necessary.  Bold expression are not used at table 1 and table 2. 

Author Response

Response to Reviewer 2

This article reported the association of the risk of falling and any related anxiety with oral-related conditions (number of teeth, occlusal support, masticatory performance, occlusal force, and tongue pressure) and physical motor functions (gait speed, knee extension force, and one-legged standing) in the elderly individuals. By a logistic regression model, some oral related conditions such as a decrease in occlusal force and left/right occlusal imbalances were related to physical motor function leading to an increase of the falling risk and anxiety of falling. In addition, the author analysed the association by sex.

The thesis of study was evaluated to be timely at the increase of the elderly population and health promotion for the elderly. The study employed the proper statistical tests. The conclusions of the study were derived from the study results.

However, some minor corrections are required. Bold expressions of table 3, table 4 and table 5 are not necessary.  Bold expression are not used at table 1 and table 2.

ž   Thank you for your review and for the kind comments. Tables 3, 4, and 5 have been revised according to your suggestions.

Round 2

Reviewer 1 Report

The authors seem to address all the point raised. Well done.